# Housing Affordability in the United States: Price-to-Income Ratio by Pareto Distribution

Francisco Vergara-Perucich 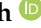

Nucleo de Investigación Centro Producción del Espacio, Universidad de Las Américas, Santiago 7500975, Chile; jvergara@udla.cl

**Abstract**

This study integrates the price-to-income ratio (PIR) with Pareto distribution characteristics to provide a novel approach for evaluating home affordability across U.S. counties. The methodology offers a new lens for the analysis of home affordability by capturing both the extreme values and central tendencies of PIR. The study normalizes the resulting Pareto parameters to a common scale and integrates data from the Zillow Home Value Index and the U.S. Department of Commerce's SAIPE program to create a single affordability index. The findings point to significant regional differences: coastal and urban regions, such as California and New York, face significant affordability challenges, whereas the Midwest, especially Kansas, has higher affordability. The results highlight the significance of targeted policy interventions and are consistent with the body of research on systemic risk and housing market dynamics. This study also opens new avenues for future research, including the impact of economic factors on affordability and cross-regional comparative studies. The suggested approach encourages more equitable access to housing by providing policymakers with a useful tool to track and manage challenges related to housing affordability.

**Keywords:** Pareto; United States of America; housing affordability; price-to-income ratio

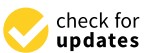

## 1. Introduction

The real estate crisis of 2007 in the US offered many lessons about economics and policy. The crisis showed how interconnected the housing market is with the broader economy, with the downturn leading to a drop in residential investment, consumption, and employment [1]. Real estate collapses often coincide with economic recessions, higher unemployment, and falling output [2], reinforcing that the housing sector is critical to overall economic health and financial stability [3]. A key lesson was that easier access to financing does not guarantee long-term affordability; while it can boost short-term homeownership, it can also inflate property values, encourage speculative purchases, and increase default risk [1,4]. Factors precipitating the 2008 bubble included financial deregulation and the widespread origination of subprime mortgages [5]. The rising cost of housing and the speculative frenzy surrounding its brittle nature—which sprang from a variety of social relation circumstances in the market—fueled the bubble. Evidence suggests bubble-like conditions in U.S. home prices existed both before 2008 and after 2013 [6].

The health of housing markets can be assessed through the distribution of home values, as rising price disparities can indicate housing bubbles [7]. This is particularly relevant as behavioural factors can sustain speculative bubbles; for instance, a survey during a housing bubble in Beijing found that "pluralistic ignorance"—where individuals mistakenly believe

their own view is not the majority view—caused continued speculative buying despite a widespread awareness of excessive prices [8]. The monitoring of metrics like price–rent discrepancies is thus crucial for predicting crises that may affect the entire economy. Decreasing housing affordability is strongly associated with higher systemic financial risk, a link observed internationally. In the UK, for instance, an unsustainable rise in house prices amplified risk across the financial system [9], and broader studies have found that riskier bank loan portfolios are often associated with worse housing affordability and higher family debt [10]. Beyond macroeconomic stability, housing affordability is a cornerstone of sustainable development. The United Nations' Sustainable Development Goal 11 (SDG 11) aims to 'ensure access for all to adequate, safe and affordable housing' [11,12]. The persistent affordability crisis in the U.S. thus represents a direct challenge to achieving these global sustainability targets, underscoring the need for robust metrics that can accurately diagnose the problem at a granular level.

*1.1. Measuring Housing Affordability: A Review of Common Metrics and Their Limitations*

The evaluation of housing affordability has evolved beyond simple income-to-expenditure ratios, with researchers exploring a variety of multidimensional approaches. The most common metric is the housing cost burden or affordability ratio, which typically defines unaffordability as housing costs exceeding a fixed 30% of household income. While widely used, this ratio-based approach is criticized for its failure to account for other essential non-housing expenses, which can lead to a significant underestimation of financial hardship, particularly in economically distressed areas [13].

In response to these limitations, the residual income approach was developed to assess whether a household has sufficient income for non-housing necessities after paying for shelter. Studies show that residual income measures often better capture the hardships faced by households [14] and more closely reflect subjective perceptions of affordability [15]. However, this method also faces conceptual challenges, as outcomes can vary significantly based on how basic budget norms are defined [16]. As an alternative, the Minimum Income Standard (MIS) has been proposed to provide a more robust assessment of need, though it has limitations regarding its applicability to diverse family types and its sensitivity to periodic updates [17].

The academic debate reflects the concept's complexity, with affordability shifting from a purely social policy concern to a broader urban challenge affecting low- and middle-income households [18]. This has led to explorations of short-term versus long-term affordability [19] and the critical relationship between housing affordability and residents' physical and mental health [20]. Despite this progress, researchers criticize conventional methods for neglecting sustainability characteristics [21] and note that challenges remain in bridging the gap between measuring housing affordability and ensuring the availability of affordable housing [22]. Ultimately, the choice of metric profoundly influences the perceived scale of the affordability crisis, with different measures yielding a broad range of estimates [23]. While these metrics are invaluable for assessing household-level financial stress, they share a common limitation: they often rely on fixed thresholds and are not designed to capture the full distributional nature of affordability across a market. This focus on central tendency can mask the risk of extreme price-to-income events, which are critical for understanding market instability and systemic risk [9].

*1.2. The Spatial Dimension of Affordability and the Rationale for a Distributional Approach*

There is widespread agreement that the U.S. is facing a housing affordability crisis, especially for low-income households [24–26]. The core issue is that too many households allocate an unsustainable portion of their income to housing. This is driven by competing factors: in some locations, particularly coastal cities, restrictive rules and supply shortages

have caused housing costs to soar [25], while in others, a lack of sufficient income is the primary barrier [27]. The crisis, however, is not geographically uniform. The impacts of the 2008 housing crisis were remarkably disparate across regions, shaped by local variables like urban layout, lending practices, and market integration [28,29]. In the aftermath, while some argue the systemic factors of the 2007 crisis no longer exist [30], the crash fundamentally upended a generation's belief in steadily rising property values [31].

This profound spatial heterogeneity is not unique to the United States. International research has documented significant affordability gaps and unequal spatial patterns across cities in Europe [32,33], developing Asia [34], and within advanced economies like the UK [35,36], Australia [37], and Canada [38]. These studies confirm that housing affordability is fundamentally a spatial issue, demanding analytical tools that can effectively capture geographic variation. This geographic and temporal complexity, coupled with the methodological limitations of traditional metrics, underscores a critical research gap. Standard measures often fail to capture the full picture, especially the distributional extremes and spatial variations critical for assessing systemic risk.

This manuscript introduces a methodological experiment that directly addresses this gap by combining the price-to-income ratio (PIR) with Pareto distribution parameters. The use of the PIR is crucial in urban economics; for example, studies in China have analysed its spatial–temporal evolution, finding that it follows a normal distribution there [39]. This study proposes a different path, analysing the PIR not as a simple median, but as a full distribution to conduct a market-wide analysis of tail-risk—the probability of extreme, high-impact events. Using Pareto distributions to model the PIR across U.S. counties, it is possible to assess affordability risk based on the frequency of extreme values. This integration represents a notable advance, offering a unique approach, not found in the current literature, to capture both central tendencies and extreme values, moving beyond traditional metrics limited to basic ratios.

## 2. Materials and Methods

### 2.1. Pareto Distribution Rationale

According to their size or magnitude, occurrences' frequencies are described by Pareto distributions, a type of statistical distribution [28,29]. A mathematical property known as "scale invariance", which denotes that the distribution retains its shape regardless of the scale at which it is observed, characterizes these distributions. Power-law distributions can shed light on how housing costs are distributed among various locations or regions in the context of housing pricing. The Pareto distributions can be applied to a wide range of topics related to urban studies, geographic research, and urban economics. For example, they have been used to measure city size distributions [30], discuss population in different central areas of cities [31], and analyse the value of time in transportation [32]. The power-law relationship describes the relationship between an event's size and frequency in a Pareto distribution. Applying a Pareto distribution to the price-to-income ratio (PIR) and home affordability can assist in identifying places where housing affordability is particularly low as well as estimate the risk of high PIR values. For this research on studying housing affordability, the assessed concepts are summarized in Table 1.

Price-to-income ratio (PIR) comparisons between counties are analysed in order to calculate the affordability risks associated with the property market. The Pareto distribution, which offers insights into the distribution's properties including the likelihood of extreme values and the general imbalance in housing affordability, was fitted to the PIR data in this study in order to achieve this. This study specifically employs the Pareto distribution due to its strength in modelling the 'tail' behaviour of distributions, which is critical for affordability analysis. While other distributions like the log-normal or Weibull can also model

skewed data, the Pareto distribution is theoretically grounded in describing phenomena where a small number of events account for a large portion of the outcomes—in this case, where a fraction of properties have extremely high PIRs that disproportionately signal market stress. The focus is not on modelling the entire PIR distribution, but specifically on quantifying the prevalence and threshold of these extreme values, which represent the most acute affordability challenges. This approach is conceptually aligned with the use of Pareto and power-law distributions in urban studies to analyse phenomena with inherent inequalities, such as city size distributions, making it a pertinent choice for exploring the extremes of housing unaffordability.

**Table 1.** Pareto distribution factors for housing affordability by PIR.

| Aspect | Description |
| --- | --- |
| Extreme Values | A Pareto distribution describes situations where extreme values occur more frequently than in a normal distribution. In housing affordability, extreme PIR values indicate highly unaffordable housing, signalling potential risks. |
| Assessing Risk | Fitting a Pareto distribution to PIR data estimates parameters $\alpha$ and $x_{min}$. Larger $\alpha$ values mean less frequent extreme events, while smaller values suggest higher likelihood of extremes. |
| Identifying High-Risk Areas | Parameters from the Pareto distribution help identify areas with high risk of unaffordable housing. Counties with PIR values in the tail (larger than $x_{min}$) face significant housing affordability concerns. |
| Policy Implications | Insights from the Pareto distribution guide policies to address housing affordability. Strategies like increasing affordable housing supply or offering financial assistance target areas with long-tailed distribution. |

It is important to clarify the specific contribution of this method to spatial analysis. The approach is not designed to formally model spatial dependence or quantify spillover effects between counties, as would be the case in spatial econometric models. Instead, its primary spatial contribution is to first characterize the internal risk profile of each geographic unit and then to map the spatial distribution of these risk profiles. By moving beyond a single measure of central tendency (like median PIR), the method can differentiate between a county where unaffordability is uniform and one where it is driven by a heavy tail of extreme values. Visualizing the composite indicator, therefore, reveals the geographic clustering of market types—identifying regions that share similar systemic affordability stresses, a spatial pattern that a map of median values would obscure. In this sense, the analysis serves as a critical diagnostic step, providing the granular, distribution-aware data necessary to identify regions of high tail-risk where more formal spatial-dependence models could be most fruitfully applied.

### 2.2. Justification for the Pareto Distribution

Price-to-income ratio (PIR) comparisons between counties were analysed to calculate the affordability risks associated with the property market. To this end, the Pareto distribution was fitted to the PIR data. This study's choice of the Pareto distribution is theoretically driven by its strength in modelling the "tail" behaviour of distributions, which is critical for affordability analysis. The objective is not to model the central tendency of the entire PIR distribution but to specifically quantify the prevalence and threshold of extreme values, which represent the most acute affordability challenges.

While other distributions like the log-normal or Weibull can also model skewed data, the Pareto distribution is the canonical model for power-law relationships. These describe phenomena where a small number of events—in this case, homes with exceptionally high PIRs—are far more frequent than in a normal distribution and disproportionately signal

market stress. This application does not claim that the model is the best fit for the entire PIR dataset, but rather that it is the most appropriate tool for analysing the power-law dynamics in the upper tail. This approach is conceptually aligned with the use of Pareto distributions in urban studies to analyse phenomena with inherent inequalities, such as city size distributions.

The Pareto distribution is defined by its probability density function (PDF):

$$P(x) = \frac{\alpha \times x_{min}^{\alpha}}{x^{\alpha+1}}$$

where $P(x)$ is the probability density of an event with size $x$; and $\alpha$ is the shape parameter, indicating how quickly the distribution tails off. $x_{min}^{\alpha}$ is the minimum possible value of $x$ in the distribution. A high shape parameter $\alpha$ implies a steep decline in probability for larger values, suggesting lower likelihoods of extreme values, while a low $\alpha$ indicates a heavier tail, with higher probabilities for extreme values. The process was conducted in R 4.5, using the code presented in Appendix A for calculating the probability density function, for the period between 2016 and 2021, as most counties had more information for running the study in this period (Appendix A). The geographical scale is at the county level.

While several distributions can model skewed data, this study's choice of the Pareto distribution is driven by its focus on the 'tail behaviour' of the price-to-income ratio (PIR). The objective is not to model the central tendency but to specifically quantify the characteristics of extreme unaffordability. The Pareto distribution is the canonical model for power-law relationships, which describe phenomena where a small number of events (in this case, homes with exceptionally high PIRs) are far more frequent than in a normal or lognormal distribution. Applying this model is not a claim of it providing the best fit for the entire PIR dataset, but rather that it is the most appropriate tool for analysing the power-law dynamics in the upper tail, which is the locus of the most severe affordability challenges and systemic risk. This aligns with its application in urban studies for phenomena characterized by inherent inequality and extreme outcomes

### 2.3. Parameter Estimation

The process was conducted in R, utilizing the poweRlaw package, a standard tool for fitting heavy-tailed distributions. For each county's PIR data, the parameters of the Pareto distribution were estimated independently. The $x_{min}$ parameter, representing the threshold above which the power-law behaviour holds, was estimated by minimizing the Kolmogorov–Smirnov (KS) statistic between the data and the fitted model. Subsequently, the shape parameter $\alpha$ was estimated using the maximum likelihood estimation (MLE) method for all data points greater than or equal to the estimated $x_{min}$. A formal goodness-of-fit test was not performed as part of this initial analysis, which should be noted as a limitation; however, these estimation methods are standard practice for robustly fitting power-law distributions. After the calculation of the values, the 'group_by' and 'group_modify' functions were used to apply the 'fit_pareto_for_geoid' function to each GEOID, which represents USA counties (Figure 1). GEOID is a standard geographic identifier code used by the U.S. Census Bureau to represent counties. This step ensures that each county's PIR data is processed independently to fit the Pareto distribution. The estimated parameters $\alpha$ and $x_{min}$ are interpreted based on their ranges. For $\alpha$, the following criteria are used:

- Low alpha: Indicates a heavy-tailed distribution with a higher likelihood of extreme PIR values, suggesting significant inequality in housing affordability.
- Medium alpha: Suggests moderate inequality with a balance between common and extreme PIR values.

- High alpha: Implies a steep decline in the distribution's tail, meaning extreme PIR values are rare, indicating lower inequality.

  For $x_{min}$:

- Low $x_{min}$: The power-law behaviour starts at very low PIR values, suggesting broad applicability of the Pareto distribution across low PIR values.
- Medium $x_{min}$: Indicates that a significant portion of the data follows the distribution, but not the very low PIR values.
- High $x_{min}$: The power-law behaviour starts at higher PIR values, meaning that only the higher PIR values follow the distribution.

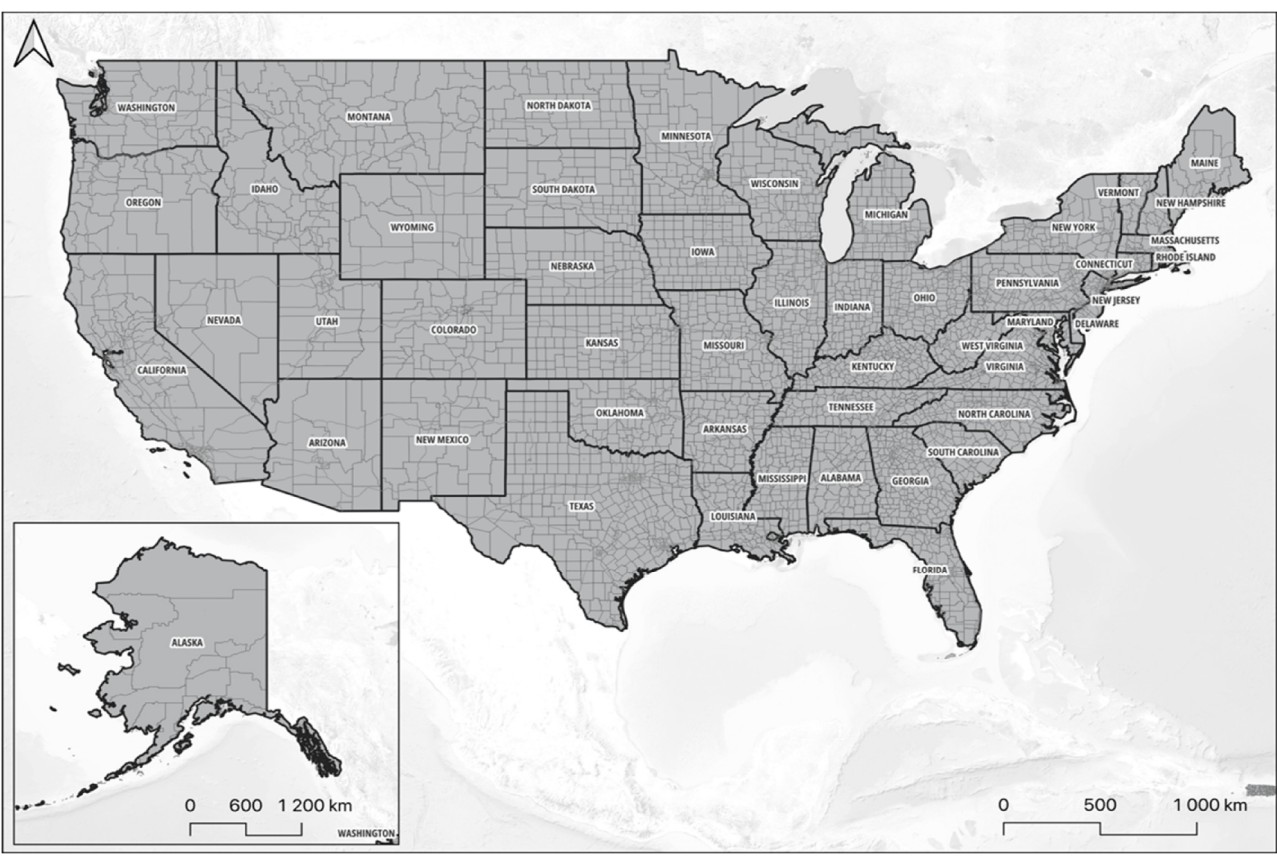

**Figure 1.** USA counties analysed. The geographic boundaries of the U.S. counties included in the analysis. This map illustrates the spatial units for which the price-to-income ratio and Pareto parameters were calculated.

In order to help to understand the results, a unique indicator was produced. This indicator integrates both $\alpha$ and $x_{min}$ into a single value by normalizing both values. This creates a more common scale for then combining them. The combined indicator reflects the significance of each parameter. In this method, a unique indicator is created by normalizing the shape parameter $\alpha$ and the minimum value $x_{min}$ of a Pareto distribution to a common scale, ranging from 0 to 1. First, the minimum and maximum values for $\alpha$ and $x_{min}$ are defined. Then, a normalization function is used to scale each value to the [0, 1] range. The normalized values are then combined into a single indicator by averaging them. This combined indicator integrates the influence of both parameters, facilitating comparative analysis across different regions or datasets. The result is stored in a dataframe, adding columns for the normalized values and the combined indicator, which can then be used for further analysis or interpretation. This approach facilitates the further visualization of the results.

*2.4. Data Sources*

The calculation of the price-to-income ratio was based on two sources: the median household income per county from the USA-SAIPE program and housing prices from the Zillow Home Value Index (ZHVI). The dataset utilized for analysing yearly weighted median household income across U.S. states and counties from 2000 to 2022 was sourced from the U.S. Department of Commerce, Bureau of the Census, under the Small Area Income and Poverty Estimates (SAIPE) Program. This dataset provides comprehensive and annual estimates of median household income, crucial for geographic and socioeconomic analyses. Definitions of rural classifications are accessible on the USDA Economic Research Service webpage within the 'Rural Economy & Population' section. Prepared by the USDA Economic Research Service, this data is current as of 16 June 2023. The Zillow Home Value Index (ZHVI) is a widely recognized metric that indicates the "typical" home value in a specific area, such as a metro region, city, or ZIP code. It is calculated as a weighted average of the middle third of homes in the region. The ZHVI is available in different formats, including smoothed and seasonally adjusted versions for consumer presentations and raw versions for real-time market assessments. This index provides both historical data and forecasts, making it useful for analysing home value trends and growth rates over time. To differentiate itself from median or actual selling prices, the Zillow Home Value Index (ZHVI) shows the "typical" home value in a given area. It is computed as a weighted average of the middle third of properties. By reducing short-term market volatility, this seasonally adjusted and smoothed index provides a more consistent long-term picture of housing trends. The ZHVI is not without restrictions, though, especially when it comes to excluding real-time price variations, because its smoothing procedure can mask sudden changes in the market. Although the raw ZHVI responds to shifts faster, its higher volatility causes noise to be introduced. Furthermore, the ZHVI may miss important affordability issues at the extremes by concentrating on the middle third of the market, especially in the high-end or low-end housing sectors.

## 3. Results

Figure 2 illustrates the results of this analysis: the price-to-income ratio (PIR) across U.S. counties. The visual representation reveals significant disparities in housing affordability across different regions. Counties with lower combined indicators, predominantly found in the Midwest, exhibit higher affordability. These regions have lower PIR values, indicating that households spend a manageable portion of their income on housing. The stable housing markets in these areas, with lower prices relative to incomes, reflect a balanced demand–supply equation, making housing accessible to a broader segment of the population. Conversely, counties with higher combined indicators, often located in coastal and metropolitan areas, display severe affordability challenges. These regions show elevated PIR values and significant variability in housing prices, indicating that a substantial portion of household income is allocated to housing costs. High-demand urban centres like those in California, New York, and Massachusetts exemplify this trend, where housing prices have escalated.

Table 2 highlights the housing affordability in the ten most affordable U.S. counties, evaluated using the price-to-income ratio (PIR) and Pareto distribution parameters $\alpha$ and $x_{min}$. These parameters are normalized to a common scale (0 to 1) and combined into a single affordability indicator. This composite measure reflects the affordability of housing in each county, with lower values indicating higher affordability. For instance, Hodgeman County, Kansas, exhibits an $\alpha$ of 4.5274 and an $x_{min}$ of 0.3856, normalized to 0.0087 and 0.0070, respectively, resulting in an overall indicator of 0.0079. This suggests highly affordable housing conditions. Similarly, Rawlins County, Kansas, has an affordability indicator

of 0.0080, with normalized values of 0.0084 for $\alpha$ and 0.0076 for $x_{min}$. These counties demonstrate stable housing markets with low PIR values, indicating manageable housing costs for residents.

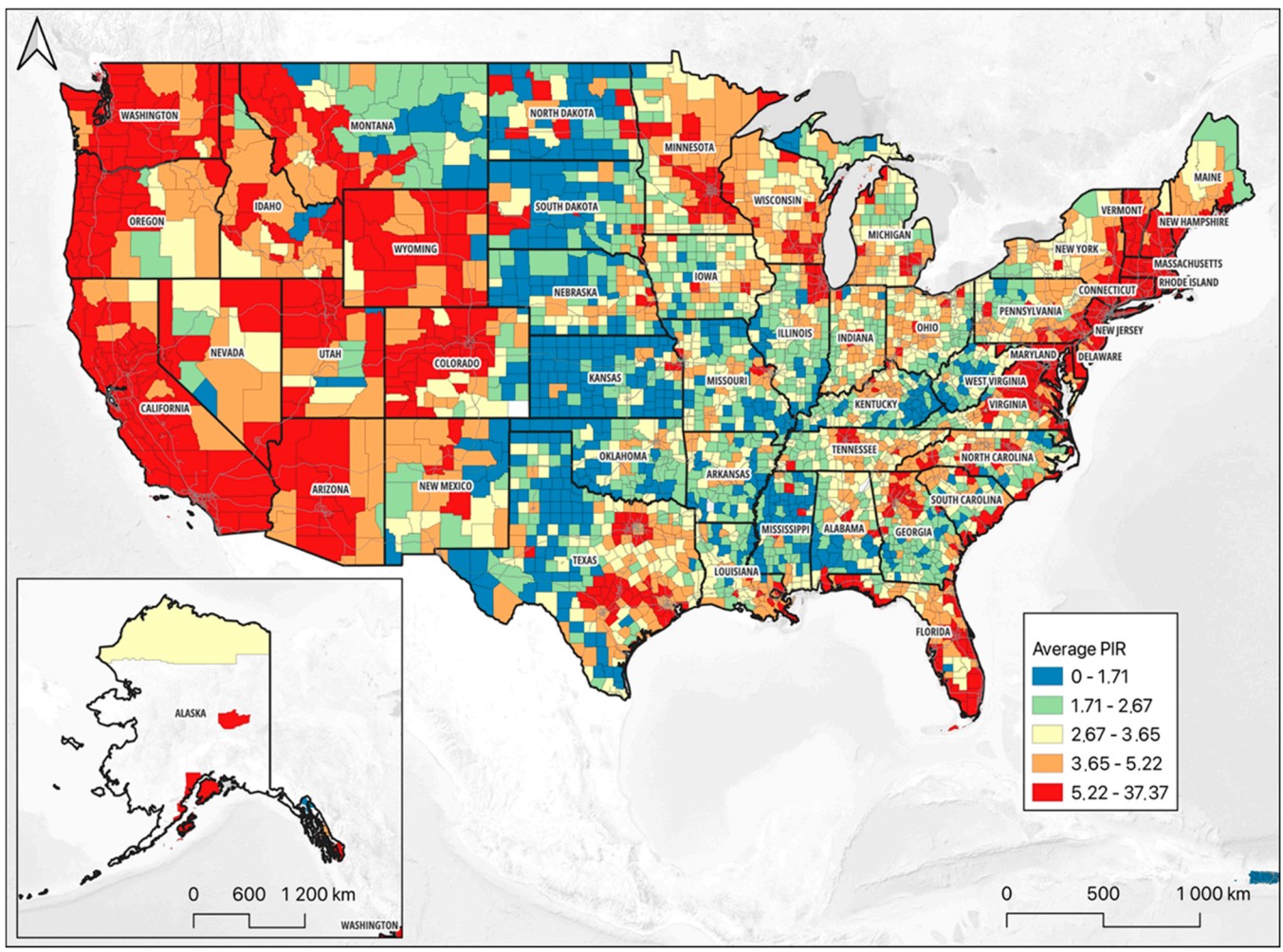

**Figure 2.** Average price-to-income ratio per county, 2016–2021.

**Table 2.** Summary of results for the 10 most affordable counties.

| County | Alpha | $x_{min}$ | Alpha (Normalized) | $x_{min}$ (Normalized) | Indicator Housing Affordability |
|---|---|---|---|---|---|
| Hodgeman County, Kansas | 4.5274 | 0.3856 | 0.0087 | 0.0070 | 0.0079 |
| Rawlins County, Kansas | 4.4331 | 0.4062 | 0.0084 | 0.0076 | 0.0080 |
| Wallace County, Kansas | 4.8426 | 0.3941 | 0.0096 | 0.0073 | 0.0084 |
| McDowell County, West Virginia | 5.0189 | 0.3993 | 0.0101 | 0.0074 | 0.0087 |
| Marshall County, South Dakota | 1.8113 | 0.7820 | 0.0009 | 0.0177 | 0.0093 |
| Woodson County, Kansas | 3.7159 | 0.6113 | 0.0064 | 0.0131 | 0.0097 |
| Stafford County, Kansas | 4.2047 | 0.5978 | 0.0077 | 0.0128 | 0.0103 |
| Clark County, Kansas | 5.8472 | 0.4719 | 0.0124 | 0.0094 | 0.0109 |
| Lincoln County, Kansas | 4.1888 | 0.6995 | 0.0077 | 0.0155 | 0.0116 |
| Sheridan County, Kansas | 3.6740 | 0.7683 | 0.0062 | 0.0174 | 0.0118 |

The results also show counties like Wallace County, Kansas, and McDowell County, West Virginia, with indicators of 0.0084 and 0.0087, respectively, reflecting their relative affordability. Marshall County, South Dakota, stands out, with a significantly lower $\alpha$ of 1.8113 but a higher $x_{min}$ of 0.7820, resulting in an indicator of 0.0093. This suggests moderate affordability with some housing cost variability. The remaining counties, predominantly in Kansas, such as Woodson, Stafford, Clark, Lincoln, and Sheridan, exhibit affordability indicators ranging from 0.0097 to 0.0118. These values indicate that, despite minor variations,

housing remains relatively affordable in these regions. The combined indicators provide a clear and comparative measure of housing affordability, emphasizing the importance of regional differences in housing markets.

Table 3 shows areas with greater affordability challenges. San Francisco County, California, exhibits a high $\alpha$ of 71.9105 and an $x_{min}$ of 35.0964, resulting in normalized values of 0.2012 for $\alpha$ and 0.9442 for $x_{min}$. This yields an overall indicator of 0.5727, highlighting severe affordability issues. Similarly, Susquehanna County, Pennsylvania, shows an affordability indicator of 0.5575, with extremely high normalized $\alpha$ and relatively lower $x_{min}$, indicating substantial housing cost burdens. Counties like San Mateo, California, and Nantucket, Massachusetts, also face significant affordability challenges, with indicators of 0.5342 and 0.5090, respectively. These counties show high PIR values and substantial variability in housing prices. New York County, New York, with an indicator of 0.4986, reflects the high cost of living and significant demand for housing. Other counties such as Santa Clara, Marin, and Alameda in California, and Honolulu in Hawaii, exhibit affordability indicators ranging from 0.3260 to 0.4878. Lafayette Parish, Louisiana, also faces affordability issues, with an indicator of 0.3633. These results underscore the severe affordability challenges in major urban centres and coastal areas, where housing costs are disproportionately high relative to incomes. It is important to note the case of Susquehanna County, Pennsylvania, which presents as an extreme outlier, with an exceptionally high $\alpha$ value (351.9757). This suggests that while its $x_{min}$ is relatively low, the probability of extreme PIR values declines extremely rapidly, a highly unusual distribution. This may be attributable to unique local economic factors or a potential data anomaly for that specific county. While this study does not investigate individual outliers, its identification highlights the sensitivity of the method in detecting atypical housing market structures.

**Table 3.** Summary of results for the 10 least affordable counties.

| County | Alpha | $x_{min}$ | Alpha (Normalized) | $x_{min}$ (Normalized) | Indicator Housing Affordability |
|---|---|---|---|---|---|
| San Francisco County, California | 71.9105 | 35.0964 | 0.2012 | 0.9442 | 0.5727 |
| Susquehanna County, Pennsylvania | 351.9757 | 4.3267 | 1.0015 | 0.1134 | 0.5575 |
| San Mateo County, California | 41.4441 | 35.4687 | 0.1142 | 0.9543 | 0.5342 |
| Nantucket County, Massachusetts | 7.8013 | 37.1627 | 0.0180 | 1.0000 | 0.5090 |
| New York County, New York | 98.0000 | 26.8481 | 0.2758 | 0.7215 | 0.4986 |
| Santa Clara County, California | 42.3033 | 31.9361 | 0.1166 | 0.8589 | 0.4878 |
| Marin County, California | 18.9429 | 29.8305 | 0.0499 | 0.8020 | 0.4259 |
| Lafayette Parish, Louisiana | 206.2651 | 5.3650 | 0.5852 | 0.1415 | 0.3633 |
| Alameda County, California | 32.0553 | 22.8534 | 0.0873 | 0.6137 | 0.3505 |
| Honolulu County, Hawaii | 58.3085 | 18.2597 | 0.1624 | 0.4896 | 0.3260 |

Figure 3 displays a map of the United States, illustrating the Composite Housing Affordability Indicator (CHAI) for various counties. The colour gradient on the map ranges from yellow to dark blue, representing different levels of housing affordability. For visualization purposes, the continuous values of the composite indicator were classified into six categories using the natural breaks (Jenks) method to effectively group similar values and highlight spatial patterns.

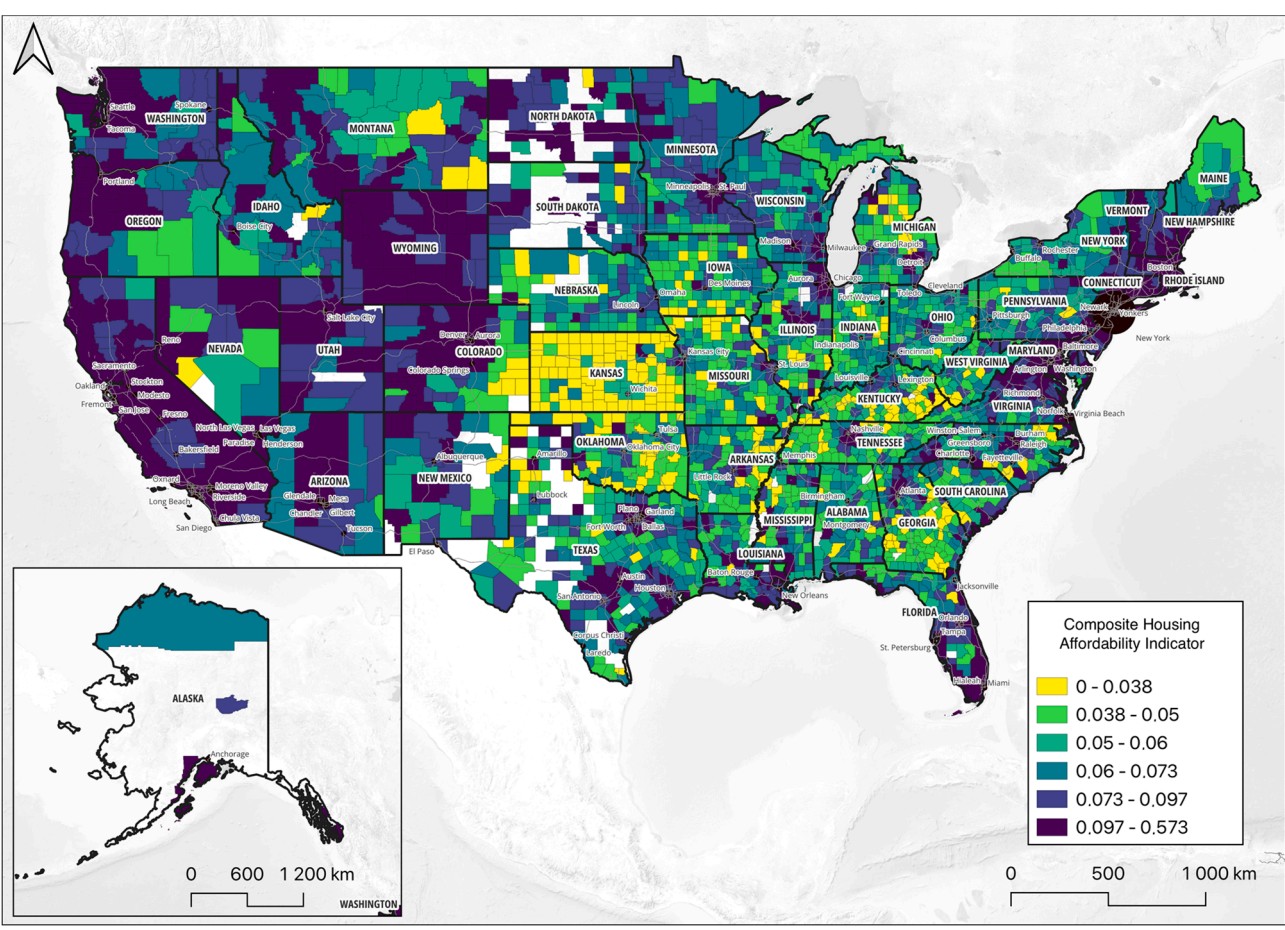

**Figure 3.** Indicator of housing affordability per county, 2016 – 2021.

Yellow: Indicates the highest affordability, with CHAI values ranging from 0 to 0.038. Counties shaded in yellow are areas where housing is most affordable, suggesting that a lower percentage of household income is spent on housing costs.

Green to Light Blue: Represents moderate affordability, with CHAI values between 0.038 and 0.06. These regions are relatively affordable but may have slightly higher housing costs compared to the yellow regions.

Dark Blue: Signifies the lowest affordability, with CHAI values between 0.073 and 0.573. Counties in dark blue experience significant affordability challenges, where a higher proportion of household income is devoted to housing costs.

Geographically, the map shows that the Midwest, particularly Kansas and surrounding states, has numerous counties with higher affordability (yellow and light green areas). In contrast, many coastal and metropolitan areas, such as parts of California, the Northeast, and certain regions in the South, display lower affordability (darker blue areas). This visual representation underscores the significant variation in housing affordability across the United States, highlighting regions where housing costs are a substantial burden on residents' incomes. The map provides a clear and comparative view of housing affordability, aiding policymakers and researchers in identifying areas with critical housing affordability issues, providing evidence to formulate targeted interventions to address these challenges.

## 4. Discussion

Based on significant regional differences that are consistent with previous research on the topic, this study's findings highlight the complicated picture of housing affordability in the United States. A strong framework for evaluating housing affordability across U.S.

counties has been introduced with the price-to-income ratio (PIR) and the Pareto distribution parameters $\alpha$ and $x_{min}$. This methodological approach is essential to comprehending both the central trends of property prices in relation to income and the extremes that indicate substantial financial constraints on individuals. Moreover, the analytical approach can be replicated for additional analyses.

The use of Pareto distributions sheds additional light on the spatial heterogeneity in housing affordability that the results reveal. Housing affordability is particularly troublesome in locations with extreme PIR values, which are identified by capturing the heavy tails of the distribution. For policymakers looking to properly focus initiatives, this is essential. A clear, comparable measure of affordability is provided by the use of normalized parameters $x_{min}$ and $\alpha$ and their integration into a single indicator, which makes focused policy-making easier. While this study presents these spatial disparities through descriptive mapping, the proposed indicator inherently captures spatial variation in a way traditional metrics cannot. By fitting a Pareto distribution to each county, the study quantifies two key aspects of local inequality: the severity of extreme unaffordability (the shape parameter $\alpha$) and the PIR level at which this extreme behaviour begins ($x_{min}$). A low $\alpha$ in a county signifies a greater probability of encountering homes with exceptionally high PIRs compared to another county with the same median PIR but a higher $\alpha$. Therefore, the mapping of the composite indicator is not merely descriptive; it is a visualization of the geographic distribution of tail-risk in housing affordability, identifying clusters of counties that face similar systemic affordability stresses. While a formal spatial econometric analysis could quantify the degree of this clustering, the method provides the granular, distribution-aware data necessary for such an analysis.

The regional patterns identified in the results, particularly the acute unaffordability on the coasts and relative affordability in the Midwest, align with established research on the structural drivers of housing crises. To move beyond a descriptive comparison, it is feasible to substantively test the alignment of the findings with frameworks like that of Bangura and Lee [37], which suggest severe unaffordability is rooted in factors like restrictive zoning, supply constraints, and speculative investment. An exploratory analysis of the most extreme cases from the results provides strong empirical support for this connection. For instance, San Francisco County, California, and Nantucket County, Massachusetts—ranked among the least affordable in Table 3—are prime examples. San Francisco is widely documented as having some of the most restrictive land-use regulations and lengthy discretionary review processes in the nation, which severely constrain housing supply in the face of high demand. Similarly, Nantucket's extreme unaffordability is a textbook case of geographic supply constraints (as an island) combined with intense demand from a seasonal, high-income tourism and second-home market, a key form of speculative investment pressure.

In stark contrast, Hodgeman County, Kansas—the most affordable county identified in Table 2—is characterized by the near-total absence of these pressures. As a rural, agricultural county with low and declining population density, it faces minimal demand, has ample land with few restrictive zoning codes, and lacks any significant speculative investment. This alignment—where the counties the indicator identifies as affordability extremes correspond precisely to the presence or absence of the structural drivers highlighted by Bangura and Lee [37]—provides empirical weight to the findings. It suggests the Pareto-based indicator is not just mapping prices but is effectively capturing the real-world outcomes of these underlying market-shaping forces.

This dynamic is reflected more broadly in the Midwest, where residences are frequently owned by the same families for generations, particularly in agricultural settings. This legacy ownership lowers price-to-income ratios by easing the burden of mortgages and

curbing speculative housing price hikes. Coupled with low population density and limited employment prospects, this historical background suggests that the affordability of these areas is impacted by both balanced housing markets and the stagnation of housing demand. Because there is less pressure on the housing supply, fewer new inhabitants are attracted by the lack of favourable economic prospects, which helps to maintain lower housing costs.

Conversely, the comparatively greater cost of housing in remote Western states like Wyoming and Colorado points to a distinct dynamic. Since ranching and outdoor recreation place a high value on land, these areas frequently serve these industries, which drives up the cost of housing. Land-use patterns driven by conservation and recreation economies constrain development and increase land values, reflecting a different set of structural pressures on affordability. Rising demand due to growing interest in rural living and tourism in these locations may also cause costs to rise in ways that defy conventional affordability models used for other rural regions.

Accordingly, Figure 3 illustrates how historical ownership patterns, population trends, and regional economic activity interact to shape housing affordability across U.S. counties, indicating that affordability measurements need to take these localized economic realities into consideration. The stark divergence between coastal and Midwestern counties, as revealed by the indicator, contributes to the broader academic debate on urbanization and spatial inequality. The high tail-risk identified in coastal metropolitan areas aligns with theories of 'superstar cities', where global capital flows and high-wage job concentration create winner-take-all housing markets. From a sustainability perspective, such extreme unaffordability undermines the social pillar of sustainable development by fostering exclusion and long commutes, placing further strain on urban systems. The findings thus provide a granular, distribution-aware lens through which to view these macro-level debates.

The findings serve to quantify the significant spatial heterogeneity of the relationship between household earnings and home prices. The results demonstrate how varied local economic conditions across the U.S. produce dramatically different affordability outcomes, highlighting that a national-level understanding of affordability is insufficient without a granular, county-level analysis of how these fundamental factors interact. The finding of spatially clustered unaffordability on the coasts is consistent with research linking unaffordability to systemic risk. This study advances this understanding by demonstrating that these regions are characterized not just by high median PIRs, but by heavy-tailed distributions (low $\alpha$), suggesting the risk is concentrated in the prevalence of extreme price-to-income outliers.

A key contribution of this research is the introduction of a measurement method that moves beyond central tendency (e.g., median PIR) to characterize housing affordability through its distributional properties. While the resulting geographic patterns may appear broadly similar to those derived from traditional metrics, the Pareto-based indicator reveals critical information that median-based approaches inherently miss. The primary innovation is the ability to quantify the local risk of encountering extreme unaffordability. To illustrate, consider two hypothetical counties that both have a median PIR of 7, making them appear equally unaffordable by conventional standards. However, the method might reveal that County A has a low shape parameter $\alpha$ (e.g., 2.5), while County B has a high $\alpha$ (e.g., 5.0). The low $\alpha$ in County A indicates a 'heavy tail', meaning the probability of encountering a home with a PIR of 15, 20, or higher is significantly greater than in County B. This 'tail risk' is a critical feature of a housing market, signalling potential for speculative bubbles and extreme financial strain on a segment of the population. A simple median metric is blind to this distinction. By integrating both the threshold of extremity ($x_{min}$) and the tail-heaviness ($\alpha$), the composite indicator provides a more complete picture of affordability, capturing not just the 'typical' condition but also the 'worst-case' risk profile of a local housing market. It

is this ability to differentiate between types of unaffordability that substantiates the value of the proposed method.

This allows for further research towards a more sophisticated understanding of housing affordability in all U.S. counties through the application of a methodological approach that combines Pareto distribution parameters and PIR. The results demonstrate notable regional differences, with the Midwest having greater affordability and coastal regions having serious difficulties. The findings are consistent with previous research, highlighting the intricate relationship that exists between earnings, housing costs, and economic stability. In order to help policymakers successfully address the problem of housing affordability, this study provides a clear, comparable measure of affordability. Targeted actions are made easier by the integration of normalized parameters into a single indicator, which encourages equitable access to affordable housing in all areas.

## 5. Conclusions

This study presents a novel approach to examining housing affordability by integrating the price-to-income ratio (PIR) with Pareto distribution parameters. Through the use of this framework, the research outcomes highlight the complex terrain of housing affordability in the United States, revealing significant regional disparities. The findings indicate that housing costs are more affordable in the Midwest, especially in Kansas, while coastal and urban regions like California and New York face significant affordability challenges. By using $\alpha$ and $x_{min}$, the method provides a comprehensive view of affordability, capturing both central tendencies and extreme cases. The integration of these elements into a single measure represents a significant step forward in the assessment, facilitating cross-regional comparison and adding a valuable new layer to existing research. By mapping affordability across U.S. counties, the study demonstrates how housing costs and incomes vary geographically, offering useful insights for both researchers and policymakers.

The composite indicator developed in this study is more than a measurement tool; it is a diagnostic that can be used to prioritize and tailor policy interventions. By disaggregating the indicator into its constituent parameters, policymakers can move beyond one-size-fits-all solutions and design responses that target the specific nature of a county's affordability challenge. It is possible to identify at least three distinct market profiles that call for different strategies:

Profile 1: The "Heated Market" (high $x_{min}$ and low $\alpha$). This profile represents the most severe affordability crisis, where a high floor for unaffordability is compounded by a heavy tail of extreme values. This suggests a market suffering from both systemic scarcity and intense speculative pressure. A dual-pronged policy approach is necessary, combining supply-side interventions (e.g., promoting inclusionary zoning, streamlining permits for affordable developments) with demand-cooling measures (e.g., speculation taxes, taxes on vacant properties or foreign investment) to stabilize the market.

Profile 2: The "Systemic Scarcity" Market (high $x_{min}$ and high $\alpha$). In this case, the high floor ($x_{min}$) indicates a broad-based lack of affordable housing stock, but the steep tail (high $\alpha$) suggests that extreme speculation is not the primary driver. The affordability problem is more uniform and less volatile. The policy priority here should be almost exclusively on increasing the overall supply of affordable housing through direct public investment, expanding housing choice vouchers, and creating incentives for the construction of low- and middle-income housing.

Profile 3: The "Emerging Risk" Market (low $x_{min}$ and low $\alpha$). This profile acts as an early warning signal. While the market may appear generally affordable (a low threshold for extreme values), the heavy tail (low $\alpha$) indicates growing inequality and emerging speculative bubbles at the high end. The appropriate response is preventative and tar-

geted, focusing on policies that protect entry-level buyers (e.g., down-payment assistance programs) and "taming the tail" with early-stage demand-side management before the affordability problem becomes systemic.

This nuanced, parameter-driven diagnostic demonstrates the significant added value of the proposed methodology. Additionally, the research emphasizes the importance of regularly assessing housing affordability to enable policymakers to respond quickly to emerging problems and preserve housing market stability. By providing a more sophisticated tool for identifying areas of acute affordability stress, this methodology can help policymakers better target interventions to advance the goals of SDG 11.

Several limitations of this study should be acknowledged, spanning its data sources, methodology, and geographic scale. The analysis relies exclusively on the price-to-income ratio (PIR), a metric that omits factors like mortgage rates and property taxes, and uses Zillow's ZHVI data, which may mask volatility at the market's extremes. Furthermore, regional differences in household composition could skew absolute PIR values. Methodologically, the Pareto fitting process lacks a formal goodness-of-fit test for each county, and the normalization scales used to create the composite indicator influence the results. While the method effectively captures extreme values, their full ramifications require deeper analysis. Finally, the study's county-level focus can obscure significant local affordability variations due to the Modifiable Areal Unit Problem (MAUP), particularly between urban and rural areas within the same county; more granular, neighbourhood-level research is needed to provide richer insights. Future work should also include a sensitivity analysis to test the robustness of the findings. For example, recalculating the composite indicator using different thresholds for defining the tail of the distribution (e.g., the top 15% or 20% of PIRs) would help confirm that the identified spatial patterns are not artifacts of the specific $x_{min}$ estimation method used here.

The study's approach provides a number of directions for future investigation. One possible research avenue is examining how various economic and demographic factors, such as employment rates and population growth, affect housing affordability in various locations. Furthermore, longitudinal research could track how affordability evolves over time in reaction to market fluctuations and economic cycles. Another promising avenue is extending the technique to other nations to provide insightful comparisons of different market structures and policy approaches. Moreover, while this study relies solely on the PIR, a crucial next step would be to apply this same Pareto-based methodology to the price-to-rent ratio as a robustness check. Such an analysis would offer complementary insights into whether unaffordability is driven more by ownership costs or by broader housing market valuations, especially in rental-dominated markets. Ultimately, this study should be viewed as a promising but preliminary methodological experiment. The framework presented here offers a valuable starting point, and its findings highlight the need for further validation and refinement in future research to fully unlock the potential of distribution-based affordability analysis.

**Funding:** The APC was funded by Núcleo de Investigación Centro Producción del Espacio, Universidad de Las Américas, Chile via PIR202427.

**Data Availability Statement:** The data presented in this study are available on request from the corresponding author. The data are not publicly available due to privacy or ethical restrictions.

**Conflicts of Interest:** The author declares no conflicts of interest.

# Appendix A

```r
library(dplyr)
library(tidyr)
library(poweRlaw)

fit_pareto_for_geoid <- function(data) {
    pir_long <- data %>%
    pivot_longer(cols = starts_with("PIR_"),
                 names_to = "Year",
                 values_to = "PIR") %>%
    filter(!is.na(PIR) & PIR > 0)
  if (nrow(pir_long) < 2) {
    return(tibble(GEOID = unique(data$GEOID), alpha = NA, xmin = NA))
  }
    m <- conpl$new(pir_long$PIR)
  est <- tryCatch({
    estimate_xmin(m)
  }, error = function(e) {
    return(NULL)
  })

  if (is.null(est)) {
    return(tibble(GEOID = unique(data$GEOID), alpha = NA, xmin = NA))
  }

  m$setXmin(est)
  est <- tryCatch({
    estimate_pars(m)
  }, error = function(e) {
    return(NULL)
  })

  if (is.null(est)) {
    return(tibble(GEOID = unique(data$GEOID), alpha = NA, xmin = NA))
  }

  alpha_hat <- est$pars
  xmin <- m$getXmin()
  return(tibble(GEOID = unique(data$GEOID), alpha = alpha_hat, xmin = xmin))
}

result <- df %>%
  group_by(GEOID) %>%
  group_modify(~ fit_pareto_for_geoid(.x)) %>%
  ungroup()
```

**Figure A1.** R script for calculating the probability density function.

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
