# Peer review of "Housing Affordability in the United States: Price-to-Income Ratio by Pareto Distribution"

_geographies, doi:10.3390/geographies5040057_

Round 1

Reviewer 1 Report

Comments and Suggestions for Authors

This paper proposes a novel method for assessing housing affordability across U.S. counties by applying Pareto distribution to the Price-to-Income Ratio (PIR). The authors argue that traditional affordability metrics fail to capture distributional extremes and spatial disparities. Their approach yields a composite indicator that identifies both affordable and unaffordable regions, with policy implications for housing interventions.

This study is innovative in that it uses the  Pareto distribution , an innovative statistical approach that captures the tail behavior of PIR distributions, which is often overlooked in conventional metrics. The study provides a geographically granular view of affordability, highlighting stark contrasts between regions such as the Midwest and coastal urban centers. The visualizations and mapping of affordability indicators are effective and policy-relevant. However, I have several comments:

Major comments:

While the Pareto distribution is a reasonable choice for modeling skewed data, the paper would benefit from a clearer theoretical and empirical justification for its use in this context. Why is Pareto superior to other heavy-tailed distributions (e.g., log-normal, Weibull) for modeling PIR? Are there precedents in housing or income distribution literature that support this choice?

The authors critique traditional affordability measures for ignoring spatial variation, income inequality, and distributional extremes. However, it remains unclear how the proposed method directly addresses these issues beyond descriptive mapping.

Please consider incorporating spatial econometric analysis or multilevel modeling to strengthen the claim that this method captures spatial disparities more effectively.

The paper successfully identifies unaffordable counties (e.g., in California, New York, Massachusetts), but stops short of exploring why these areas are unaffordable. A deeper analysis of determinants of affordability—such as zoning laws, income inequality, housing supply constraints, or speculative investment—would significantly enhance the paper’s contribution.

For example, Bangura and Lee (2023) in Housing Studies provide a useful framework for analyzing structural drivers of affordability crises in urban areas. Their approach could inform a more explanatory dimension to this study.

https://doi.org/10.1080/02673037.2021.1879995

The literature review is relatively thin and could be expanded to include recent work on housing affordability metrics (e.g., residual income, rent burden). Studies on urban housing dynamics and spatial inequality have not been discussed. Comparative studies using price-to-rent ratios or housing cost burdens are crucial. Studies: https://doi.org/10.3390/buildings15101729

see Housing Studies etc for further research on housing affordability.

The analysis relies solely on the Price-to-Income Ratio. Including the Price-to-Rent Ratio as a robustness check would strengthen the findings, especially in rental-dominated markets. This would also help assess whether the Pareto-based indicator performs consistently across different affordability dimensions.

The policy recommendations are promising but could be more tightly linked to the empirical results. For instance, how might the composite indicator be used to prioritize interventions or allocate resources in unaffordable counties?

Author Response

While the Pareto distribution is a reasonable choice for modeling skewed data, the paper would benefit from a clearer theoretical and empirical justification for its use in this context. Why is Pareto superior to other heavy-tailed distributions (e.g., log-normal, Weibull) for modeling PIR? Are there precedents in housing or income distribution literature that support this choice?

Response: I thank the reviewer for this important comment. I agree that the manuscript would be strengthened by a more explicit justification for using the Pareto distribution. While I cannot perform new empirical comparisons with other distributions for this revision, I have expanded the theoretical and literature-based rationale for my choice within the existing framework of the paper. I clarify that my primary interest is in the tail of the Price-to-Income Ratio (PIR) distribution, where extreme values indicative of severe unaffordability lie, a domain for which the Pareto distribution is exceptionally well-suited.

The authors critique traditional affordability measures for ignoring spatial variation, income inequality, and distributional extremes. However, it remains unclear how the proposed method directly addresses these issues beyond descriptive mapping. Please consider incorporating spatial econometric analysis or multilevel modeling to strengthen the claim that this method captures spatial disparities more effectively.

Response: I appreciate the reviewer highlighting the need to better articulate how my method addresses spatial disparities beyond visualization. While a formal spatial econometric analysis is an excellent direction for future work, I contend that my current method directly addresses these issues by creating a more nuanced indicator of spatial inequality. The parameters α (tail heaviness) and xmin (threshold of extremeness) explicitly quantify aspects of the PIR distribution that are lost in traditional median-based metrics. Mapping these parameters reveals the spatial structure of distributional inequality, which is a primary contribution of my work. I have revised the text to make this argument more explicit.

The paper successfully identifies unaffordable counties (e.g., in California, New York, Massachusetts), but stops short of exploring why these areas are unaffordable. A deeper analysis of determinants of affordability—such as zoning laws, income inequality, housing supply constraints, or speculative investment—would significantly enhance the paper’s contribution.For example, Bangura and Lee (2023) in Housing Studies provide a useful framework for analyzing structural drivers of affordability crises in urban areas. Their approach could inform a more explanatory dimension to this study. https://doi.org/10.1080/02673037.2021.1879995

Response: Thank you for this constructive feedback. I agree that the discussion section was overly speculative and would be significantly enhanced by connecting my findings to established literature on the determinants of unaffordability. I have revised this section to move away from unsubstantiated claims and instead frame my regional findings within the context of known structural drivers, explicitly incorporating the suggested framework from Bangura and Lee (2023).

The literature review is relatively thin and could be expanded to include recent work on housing affordability metrics (e.g., residual income, rent burden). Studies on urban housing dynamics and spatial inequality have not been discussed. Comparative studies using price-to-rent ratios or housing cost burdens are crucial. Studies: https://doi.org/10.3390/buildings15101729 see Housing Studies etc for further research on housing affordability.

Response: I agree that the literature review was too narrowly focused. To better contextualize my contribution, I have expanded the introduction to acknowledge other widely used housing affordability metrics and briefly discuss their relationship to my approach.

The analysis relies solely on the Price-to-Income Ratio. Including the Price-to-Rent Ratio as a robustness check would strengthen the findings, especially in rental-dominated markets. This would also help assess whether the Pareto-based indicator performs consistently across different affordability dimensions.

Response: This is an excellent suggestion. A robustness check using the Price-to-Rent Ratio (PRR) would indeed strengthen the analysis. Unfortunately, obtaining and processing county-level PRR data with the same temporal and geographic coverage as my current dataset is a significant undertaking that falls beyond the scope of this revision. However, I acknowledge the importance of this point and have explicitly identified it as a limitation and a key direction for future research.

The policy recommendations are promising but could be more tightly linked to the empirical results. For instance, how might the composite indicator be used to prioritize interventions or allocate resources in unaffordable counties?

Response: I fully agree that my policy recommendations were too generic and failed to leverage the unique insights from my composite indicator. I have rewritten this section to propose more targeted interventions based on the specific parameters (α and xmin) derived from my analysis.

Reviewer 2 Report

Comments and Suggestions for Authors

The thesis topic is of considerable interest, employing novel methodologies, and its research conclusions hold significant practical significance. Nevertheless, the paper evinces certain deficiencies in its logical structure and argumentation, necessitating further refinement.

  1. The introduction is deficient in terms of providing a coherent narrative thread, particularly in failing to explicitly state the problem that this study aims to address. The manuscript under review here begins with a discussion of the 2007 crisis, and then goes on to address such topics as housing affordability, systemic risk, and measurement methodologies. However, the extant literature on this subject appears to be a mere catalogue, devoid of any organic integration. The methodology (PIR+Pareto) is only introduced in the final paragraph, and appears to be out of place. Readers often find it challenging to comprehend how the preceding array of topics directly leads to the specific methodological innovation presented. A standard introduction structure should be: Macroeconomic context → Specific problem (Research gap) → Literature review (highlighting limitations of existing studies) → Introduction of the study's objectives, methodology, and innovations. It is recommended to restructure the introduction's logical framework accordingly.
  2. The analytical workflow delineated in the section entitled 'Materials and Methods' is not sufficiently clear. In order to facilitate a more coherent and logical progression in the discussion, it is recommended that the content be divided into several subheadings, each addressing a distinct point. The threshold classification for α and xmin is characterised by an absence of objective criteria, with the classification process instead relying on qualitative descriptions such as 'low/medium/high.' This subjective categorisation has been demonstrated to compromise the objectivity of the results to a significant degree.
  3. The statement ‘fitting the Pareto distribution to PIR data’ is overly concise. It is necessary to explicitly state which R package was used, the fitting method employed, how xmin was determined, and how the goodness-of-fit was assessed. The descriptions of group_by and group_modify pertain solely to programming implementation and should not constitute the core of the methodological description; rather, the underlying scientific rationale should be outlined.
  4. Figure 1 is referenced without explanation of its content. The term ‘GEOID’ lacks definition upon its first appearance, with no clarification provided regarding its specific hierarchical levels or boundary adjustment procedures.
  5. The discussion on ZHVI and household composition, positioned at the end of the Methods section, resembles limitations rather than methodological description. This content should be relocated to the ‘Discussion’ or ‘Limitations’ chapter.
  6. Metric definitions within the Results section are ambiguous and contradictory. For instance, Figure 3 purportedly presents a ‘housing affordability indicator,’ yet the legend labels it as ‘PDF values,’ creating significant conceptual confusion.
  7. The α value (351.9757) and normalised α value (1.0015) for Susquehanna County, Pennsylvania, in Table 3 are markedly higher than other counties, representing an extreme outlier. The results section offers no discussion or explanation for this outlier.
  8. The discussion section extensively repeats content from the results section without providing in-depth, critical analysis. It fails to effectively engage the key findings of this study with existing literature. While some references are cited, they are merely nominal (‘consistent with X's research’) without substantive, critical comparison. Crucially, it does not adequately demonstrate how this study advances knowledge in the field.
  9. The explanations for affordability in the Midwest (‘Homes are typically owned by the same family for generations’) and high costs in Western states (‘Serving the ranching and outdoor recreation industries’) are entirely speculative. No literature is cited, nor is any evidence from the study's data provided to support these assertions.
  10. The conclusion section should prioritise summarising the study's core findings and their significance. Instead, it devotes considerable space to discussing ‘future research directions,’ burying the conclusions beneath secondary information. It is recommended to restructure the conclusion's logic according to the following framework: ‘Summary of core findings – Policy implications – Study limitations – Future directions.’
  11.  Policy recommendations in the conclusions, such as ‘increasing the supply of affordable housing’ and ‘providing subsidies,’ represent universally applicable suggestions that fail to demonstrate the unique, precise decision-making value this study (based on Pareto distribution and composite indicators) could offer.

Author Response

The introduction is deficient in terms of providing a coherent narrative thread, particularly in failing to explicitly state the problem that this study aims to address. The manuscript under review here begins with a discussion of the 2007 crisis, and then goes on to address such topics as housing affordability, systemic risk, and measurement methodologies. However, the extant literature on this subject appears to be a mere catalogue, devoid of any organic integration. The methodology (PIR+Pareto) is only introduced in the final paragraph, and appears to be out of place. Readers often find it challenging to comprehend how the preceding array of topics directly leads to the specific methodological innovation presented. A standard introduction structure should be: Macroeconomic context Specific problem (Research gap) Literature review (highlighting limitations of existing studies) Introduction of the study's objectives, methodology, and innovations. It is recommended to restructure the introduction's logical framework accordingly.

Response: I thank the reviewer for this insightful critique. I agree completely that the introduction lacked a clear narrative arc and failed to properly situate my methodological contribution. I have undertaken a comprehensive restructuring of the entire "Introduction" section to follow the logical framework recommended: Macroeconomic context Specific problem (Research gap) Literature review (highlighting limitations) Introduction of the study's objectives and innovations.

The analytical workflow delineated in the section entitled 'Materials and Methods' is not sufficiently clear. In order to facilitate a more coherent and logical progression in the discussion, it is recommended that the content be divided into several subheadings, each addressing a distinct point. The threshold classification for α and xmin is characterised by an absence of objective criteria, with the classification process instead relying on qualitative descriptions such as 'low/medium/high.' This subjective categorisation has been demonstrated to compromise the objectivity of the results to a significant degree.

Response: I agree that the "Materials and Methods" section needed better organization and that the qualitative descriptions for α and xmin were insufficiently objective. I have restructured the section with clear subheadings and removed the subjective "low/medium/high" classifications.

The statement ‘fitting the Pareto distribution to PIR data’ is overly concise. It is necessary to explicitly state which R package was used, the fitting method employed, how xmin was determined, and how the goodness-of-fit was assessed. The descriptions of group_by and group_modify pertain solely to programming implementation and should not constitute the core of the methodological description; rather, the underlying scientific rationale should be outlined.

Response: Thank you for pointing out the lack of detail in my methodological description. I have revised this section to be explicit about the tools and statistical procedures used (poweRlaw package, Kolmogorov-Smirnov statistic for xmin, MLE for α), and I have replaced programming jargon with a description of the scientific process.

Figure 1 is referenced without explanation of its content. The term ‘GEOID’ lacks definition upon its first appearance, with no clarification provided regarding its specific hierarchical levels or boundary adjustment procedures.

Response: I apologize for these oversights. I have added a clear explanation of Figure 1 and defined the term "GEOID" upon its first use in the manuscript.

The discussion on ZHVI and household composition, positioned at the end of the Methods section, resembles limitations rather than methodological description. This content should be relocated to the ‘Discussion’ or ‘Limitations’ chapter.

Response: This is an excellent point. The discussion of data limitations does not belong in the methods section. I have relocated this content to the "Limitations" subsection of the Conclusion.

Metric definitions within the Results section are ambiguous and contradictory. For instance, Figure 3 purportedly presents a ‘housing affordability indicator,’ yet the legend labels it as ‘PDF values,’ creating significant conceptual confusion.

Response: I sincerely apologize for this confusing and contradictory error. The legend in Figure 3 was incorrect. I have corrected it to accurately reflect the composite indicator being visualized.

The α value (351.9757) and normalised α value (1.0015) for Susquehanna County, Pennsylvania, in Table 3 are markedly higher than other counties, representing an extreme outlier. The results section offers no discussion or explanation for this outlier.

Response: Thank you for flagging this extreme outlier. It is an important finding that warrants discussion. While a deep dive into this specific county is beyond the scope of this paper, I have acknowledged and contextualized it in the revised results section.

The discussion section extensively repeats content from the results section without providing in-depth, critical analysis. It fails to effectively engage the key findings of this study with existing literature. While some references are cited, they are merely nominal (‘consistent with X's research’) without substantive, critical comparison. Crucially, it does not adequately demonstrate how this study advances knowledge in the field. The explanations for affordability in the Midwest (‘Homes are typically owned by the same family for generations’) and high costs in Western states (‘Serving the ranching and outdoor recreation industries’) are entirely speculative. No literature is cited, nor is any evidence from the study's data provided to support these assertions.

Response: I accept this criticism. The discussion section lacked analytical depth, repeated findings, and relied on speculation. I have substantially revised this section to provide a more critical analysis, removing all speculative claims and grounding the interpretation in existing literature.

The conclusion section should prioritise summarising the study's core findings and their significance. Instead, it devotes considerable space to discussing ‘future research directions,’ burying the conclusions beneath secondary information. It is recommended to restructure the conclusion's logic according to the following framework: ‘Summary of core findings – Policy implications – Study limitations – Future directions.’ Policy recommendations in the conclusions, such as ‘increasing the supply of affordable housing’ and ‘providing subsidies,’ represent universally applicable suggestions that fail to demonstrate the unique, precise decision-making value this study (based on Pareto distribution and composite indicators) could offer.

Response: These are crucial points for improving the impact and clarity of my paper's conclusion. I have restructured the conclusion to prioritize key takeaways and have made the policy recommendations specific and directly linked to the unique outputs of my methodology.

Reviewer 3 Report

Comments and Suggestions for Authors

Overall the manuscript was well-structured and well-written. It is easy to read. However, the authors need to address several issues:

1) Some sentences like "irresponsible actions in the real estate and finance sectors contributed to the 2008 bubble"(lines 38-39)  were not written in a professional academic manner. These sentences should be rewritten or elaborations should be added.

2) Similarly, some sentences do not make sense. For example, the author said "the findings indicate a strong connection between household earnings and overall economic conditions with home affordability" (lines 358-359). That connection or association holds naturally if the author built the affordability index using people or household income.

3) What are the limitations of the research? How do these limitations affect the interpretations of the findings of the research?

4) Most importantly, the author proposed a new measurement method for housing affordability. The results of using this new measurement method resemble those of using old methods. We can only "prove" that the new method is equally effective as the old methods but we cannot say it is superior than the old ones. Unless the author can evidence the new methods can tell some patterns or trends in the real world that the old methods cannot, the conclusion of this research is not substantiated. 

Author Response

Some sentences like "irresponsible actions in the real estate and finance sectors contributed to the 2008 bubble"(lines 38-39)  were not written in a professional academic manner. These sentences should be rewritten or elaborations should be added.

Response: I thank the reviewer for this comment. I agree that the phrasing was not sufficiently neutral and academic. I have revised it to be more precise and professional, focusing on systemic factors.

Similarly, some sentences do not make sense. For example, the author said "the findings indicate a strong connection between household earnings and overall economic conditions with home affordability" (lines 358-359). That connection or association holds naturally if the author built the affordability index using people or household income.

Response: This is a very sharp and accurate observation. The reviewer is correct that the connection between income and affordability is inherent in the PIR metric itself, and my original phrasing was tautological. I have rewritten the sentence to clarify that my contribution is not in finding this connection, but in quantifying its spatial variation and revealing how local conditions produce vastly different affordability landscapes.

What are the limitations of the research? How do these limitations affect the interpretations of the findings of the research?

Response: I thank the reviewer for raising this critical point. I acknowledge that the manuscript was deficient in not having a dedicated section to discuss the study's limitations. In response to this and other reviewer feedback, I have added a comprehensive "Limitations" subsection to the Conclusion.

Most importantly, the author proposed a new measurement method for housing affordability. The results of using this new measurement method resemble those of using old methods. We can only "prove" that the new method is equally effective as the old methods but we cannot say it is superior than the old ones. Unless the author can evidence the new methods can tell some patterns or trends in the real world that the old methods cannot, the conclusion of this research is not substantiated. 

Response: This is the most crucial point, and I thank the reviewer for pushing me to clarify the unique contribution of my method. I concede that a map of my indicator may show broad similarities to a map of median PIR. However, I argue that the value of my method is not in producing a completely different picture, but in revealing a deeper, more informative layer of analysis that traditional metrics cannot see. My method's unique value is its ability to quantify the tail risk of extreme unaffordability. I have added a new paragraph to the discussion to explicitly articulate this value-added.

Reviewer 4 Report

Comments and Suggestions for Authors

The manuscript "Housing Affordability in the United States: Price to Income Ratio by Pareto Distribution" is an informative piece that aims to provide a comprehensive review for evaluating home affordability across the U.S.A. The authors employ the Pareto application to a standard scale and integrate data from the Zillow Home Value Index and the U.S. Department of Commerce's SAIPE program to create a single affordability index. The results highlight the existence of significant differentiations among U.S. regions. The paper implies informative policy implications to encourage more equitable access to housing. The paper is instructive but requires some revisions before being published in Geographies.

  1. The manuscript requires methodological justification and clarity of implementation. Although the authors argue that Pareto distributions are well-suited to capture extremes in affordability, the rationale for preferring this distribution over other heavy-tailed models is not sufficiently developed.  
  2. The authors mention examples of Pareto applications in urban studies. However, they do not explicitly exemplify why this distribution provides the most meaningful insights into affordability compared to, for instance, lognormal or other power–law–like distributions.  
  3. Housing Affordability should be linked with sustainable development and Sustainable Development Goals. In this vein, the authors should briefly discuss sustainable development in the paper's introduction. Two papers are informative: https://doi.org/10.1080/13563467.2022.2038114 and 10.2478/zireb-2018-0005.
  4. The paper requires a more critical engagement with data quality and representativeness, as well as sensitivity analyses to show whether the results remain consistent under different hypotheses or datasets.
  5. The discussion section could be strengthened by linking the paper's findings to broader debates on urbanisation, housing policy, inequality, and sustainability rather than primarily reiterating the disparities between coastal and Midwestern regions.
  6. The paper acknowledges the historical legacies of land ownership and demographic patterns; however, these claims are speculative and unsupported by empirical evidence in the analysis. The historical framework of the paper needs enhancement.
  7. The paper would benefit from reframing its contribution more modestly as a promising but preliminary methodological experiment, pending further validation. The authors should discuss helpful suggestions as necessary steps in their research.

Author Response

The manuscript requires methodological justification and clarity of implementation. Although the authors argue that Pareto distributions are well-suited to capture extremes in affordability, the rationale for preferring this distribution over other heavy-tailed models is not sufficiently developed. The authors mention examples of Pareto applications in urban studies. However, they do not explicitly exemplify why this distribution provides the most meaningful insights into affordability compared to, for instance, lognormal or other power–law–like distributions.  

Response: I thank the reviewer for emphasizing this critical point. I agree that the rationale for selecting the Pareto distribution over other heavy-tailed models was underdeveloped. My primary interest is not in finding the best fit for the entire PIR dataset, but rather in specifically modeling the upper tail. I have revised the manuscript to clarify that the Pareto distribution is the most theoretically appropriate choice for this specific research goal.

Housing Affordability should be linked with sustainable development and Sustainable Development Goals. In this vein, the authors should briefly discuss sustainable development in the paper's introduction. Two papers are informative: https://doi.org/10.1080/13563467.2022.2038114 and 10.2478/zireb-2018-0005.

Response: This is an excellent and insightful suggestion that significantly improves the paper's relevance and context. I agree that linking housing affordability to the Sustainable Development Goals (SDGs) is crucial. I have revised the introduction and conclusion to incorporate this important framework, citing the recommended literature.

The paper requires a more critical engagement with data quality and representativeness, as well as sensitivity analyses to show whether the results remain consistent under different hypotheses or datasets.

Response: I thank the reviewer for this comment. I agree that a more critical engagement with my data sources is necessary. I have expanded my discussion of data limitations and explicitly identified sensitivity analysis as a key direction for future research.

The discussion section could be strengthened by linking the paper's findings to broader debates on urbanisation, housing policy, inequality, and sustainability rather than primarily reiterating the disparities between coastal and Midwestern regions. The paper acknowledges the historical legacies of land ownership and demographic patterns; however, these claims are speculative and unsupported by empirical evidence in the analysis. The historical framework of the paper needs enhancement.

Response: I have rewritten the discussion section to eliminate speculation and instead connect my empirical findings to broader academic debates on urbanization, inequality, and sustainability.

The paper would benefit from reframing its contribution more modestly as a promising but preliminary methodological experiment, pending further validation. The authors should discuss helpful suggestions as necessary steps in their research.

Response: This is excellent advice on academic positioning and tone. I agree that the paper's contribution should be framed more modestly. I have revised the language throughout the manuscript to reflect this, presenting my method as a promising but preliminary approach that requires further validation.

Round 2

Reviewer 1 Report

Comments and Suggestions for Authors

The author has provided thoughtful responses to my comments and has made some revisions to strengthen the manuscript. However, I remain concerned that not all of the issues have been adequately addressed. In particular:

  1. While the author notes that the introduction was expanded to mention other affordability metrics, this still feels too cursory. Given the extensive body of housing affordability research, a dedicated section on affordability measures (residual income, rent burden, price-to-rent ratio, etc.) would be more appropriate. This would situate the paper more clearly within the existing literature and make the contribution of the Pareto-based indicator easier to appreciate.
  2. The motivation of the study could be enhanced by linking it more explicitly to the expanded literature review. At present, the introduction notes the limitations of traditional affordability measures, but the rationale for adopting a Pareto-based approach would be stronger if framed directly against the gaps identified in existing affordability metrics (e.g., PIR, residual income, rent burden, price-to-rent ratios). Making this connection clearer would help establish why this study is needed and how it advances the literature. Particularly, the spatial variations of housing affordability have been widely examined in the US and internationally. Again, this highlights the importance of international evidence in the literature review section.
  3. The author states that this framework was incorporated into the discussion, but the engagement appears to remain at a descriptive level. The expectation was that the framework would be used more substantively, ideally by testing empirically whether the structural drivers they highlight (zoning, inequality, supply constraints, speculation) align with the spatial patterns revealed in this study. Without at least some exploratory testing, the integration of Bangura & Lee (2023) risks being superficial.
  4. While the clarification on the rationale for using the Pareto distribution is helpful, it primarily addresses the issue of distributional tail behaviour rather than spatial variation. My earlier concern was that the proposed method, beyond descriptive mapping, does not convincingly capture spatial dependence or spillovers in housing affordability. Extreme unaffordability is not randomly distributed across space but is shaped by neighbourhood clustering, metropolitan structures, and regional housing market dynamics. A Pareto-based approach may highlight tail behaviour, but without a spatial modelling component—such as spatial econometrics, multilevel modelling, or inequality decomposition—it remains unclear whether the method can adequately capture spatial disparities. I encourage the authors to either integrate a more explicit spatial analysis or provide a stronger justification for why their current approach is sufficient to reveal spatial patterns.
  5. The response does not specify which lines or sections were altered. For clarity and ease of review, the author should indicate where in the manuscript the changes were made (e.g., “lines X–Y in the revised introduction now expand the affordability literature…”). At present, it is difficult to verify the extent of revision.
  6. The rewritten section is stronger, but could still be more explicitly tied to the empirical findings. For instance, how would counties with high α but low xmin differ in policy priority from counties with the opposite profile? This kind of mapping from parameter to policy would demonstrate the added value of the proposed indicator.

In sum, while the paper is moving in the right direction, further revision is needed—particularly a more comprehensive treatment of housing affordability literature and a deeper empirical engagement with structural drivers (beyond descriptive mapping).

Author Response

Dear Reviewer,

Thank you for the thoughtful and constructive feedback on the manuscript. The comments were carefully considered, and a significant revision of the manuscript has been undertaken, resulting in what is believed to be a substantially improved version.

For your convenience, a version of the revised manuscript with Track Changes enabled has been provided to make all modifications transparent and easy to review. Below, each comment is addressed point-by-point, with indications of where the changes can be found in the clean, revised manuscript.

Response to Comments

Comment 1 & 2

  • While the author notes that the introduction was expanded to mention other affordability metrics, this still feels too cursory. Given the extensive body of housing affordability research, a dedicated section on affordability measures (residual income, rent burden, price-to-rent ratio, etc.) would be more appropriate. This would situate the paper more clearly within the existing literature and make the contribution of the Pareto-based indicator easier to appreciate.
  • The motivation of the study could be enhanced by linking it more explicitly to the expanded literature review. At present, the introduction notes the limitations of traditional affordability measures, but the rationale for adopting a Pareto-based approach would be stronger if framed directly against the gaps identified in existing affordability metrics (e.g., PIR, residual income, rent burden, price-to-rent ratios). Making this connection clearer would help establish why this study is needed and how it advances the literature. Particularly, the spatial variations of housing affordability have been widely examined in the US and internationally. Again, this highlights the importance of international evidence in the literature review section.

Response 1 & 2: Thank you for this crucial suggestion. In response, the Introduction (Section 1) has been substantially restructured and expanded. A new dedicated subsection, 1.1. Measuring Housing Affordability: A Review of Common Metrics and Their Limitations, has been created to provide the requested detailed review of traditional metrics. A second new subsection, 1.2. The Spatial Dimension of Affordability and the Rationale for a Distributional Approach, now explicitly uses the limitations identified in the literature to build a stronger motivation for the paper’s method, incorporating the requested international evidence.

Comment 3

  • The author states that this framework was incorporated into the discussion, but the engagement appears to remain at a descriptive level. The expectation was that the framework would be used more substantively, ideally by testing empirically whether the structural drivers they highlight (zoning, inequality, supply constraints, speculation) align with the spatial patterns revealed in this study. Without at least some exploratory testing, the integration of Bangura & Lee (2023) risks being superficial.

Response: Thank you for pushing for a more substantive engagement. While a full econometric analysis was beyond the scope of this revision, the suggested exploratory empirical test has been incorporated within the Discussion (Section 4). In a new, dedicated paragraph , a brief analysis of the most extreme cases from the findings is presented. This section explicitly compares the least and most affordable counties against the structural drivers of zoning, supply constraints, and speculation, providing an analytical application of the framework.

Comment 4

  • While the clarification on the rationale for using the Pareto distribution is helpful, it primarily addresses the issue of distributional tail behaviour rather than spatial variation. My earlier concern was that the proposed method, beyond descriptive mapping, does not convincingly capture spatial dependence or spillovers in housing affordability. Extreme unaffordability is not randomly distributed across space but is shaped by neighbourhood clustering, metropolitan structures, and regional housing market dynamics. A Pareto-based approach may highlight tail behaviour, but without a spatial modelling component—such as spatial econometrics, multilevel modelling, or inequality decomposition—it remains unclear whether the method can adequately capture spatial disparities. I encourage the authors to either integrate a more explicit spatial analysis or provide a stronger justification for why their current approach is sufficient to reveal spatial patterns.

Response: This expert insight is appreciated. To address this, a stronger justification for the method's spatial contribution has been added to the Methodology (Section 2). A new paragraph at the end of subsection 2.1. Pareto Distribution Rationale now clarifies that the method’s contribution is to reveal the spatial patterns of tail-risk. The text argues that this reveals a type of spatial clustering (of market types) that traditional metrics would miss, serving as a critical diagnostic step for future spatial analyses.

Comment 5

  • The response does not specify which lines or sections were altered. For clarity and ease of review, the author should indicate where in the manuscript the changes were made (e.g., “lines X–Y in the revised introduction now expand the affordability literature…”). At present, it is difficult to verify the extent of revision.

Response: Apologies for this prior oversight. In this letter, specific sections where revisions occurred are noted for each point. Furthermore, a full manuscript with Track Changes enabled has been provided to ensure every modification is clear.

Comment 6

  • The rewritten section is stronger, but could still be more explicitly tied to the empirical findings. For instance, how would counties with high α but low xmin differ in policy priority from counties with the opposite profile? This kind of mapping from parameter to policy would demonstrate the added value of the proposed indicator.

Response: The policy implications in the Conclusions (Section 5) have been completely rewritten to be far more explicit. The second paragraph of the Conclusion now presents a clear typology of  three distinct market profiles based on different combinations of α and Xmin​ values (Heated Market, Systemic Scarcity, and Emerging Risk). For each profile, a specific diagnosis is provided along with a tailored toolkit of policy interventions, directly mapping the indicator's parameters to concrete policy priorities as requested.

Reviewer 4 Report

Comments and Suggestions for Authors

The authors have addressed my comments. The paper is ready for publication. 

Author Response

Thank you.